# Effects of Utilization Methods on C, N, P Rate and Enzyme Activity of Artificial Grassland in Karst Desertification Area

Shuzhen Song, Xueling Wang, Cheng He and Yongkuan Chi *

School of Karst Science, State Engineering Technology Institute for Karst Desertification Control, Guizhou Normal University, Guiyang 550001, China; 15030170031@gznu.edu.cn (S.S.); 222200171830@gznu.edu.cn (X.W.); 20010170492@gznu.edu.cn (C.H.)
* Correspondence: 201907002@gznu.edu.cn

**Abstract:** To investigate the effects of different utilization methods on the ratio characteristics of soil C, N, P and enzyme activities of grassland soil is of great significance for the sustainable utilization of grassland. In this study, the effects of different utilization methods on soil C, N, P rate and enzyme activities were investigated in artificial grasslands treated with grazing grassland (GG), mowing grassland (MG) and enclosed grassland (EG). The results showed that: (1) the contents of soil organic carbon (SOC), total phosphorus (TP) and ammonium nitrogen ($NH_4^+$-N) were EG > GG > MG; the contents of total nitrogen (TN), alkaline nitrogen (AN) and nitrate nitrogen ($NO_3^-$-N) were GG > EG > MG; the contents of available phosphorus (AP) and C:N were EG > MG > GG; the contents of C:P and N:P were MG > GG > EG; (2) β-1,4-Glucosidase (βG) and β-1,4-N-Acetylglucos-Aminidase (NAG) activities were GG > EG > MG, acid phosphatase (ACP) was EG > MG > GG, Leucine Aminopeptidase (LAP) was MG > EG > GG, C:NEEA and C:PEEA were GG > EG > MG, N:PEEA was GG > MG > EG; (3) redundancy analysis showed that AN (F = 4.90, $p$ = 0.01) was an important driver of soil enzymes. We concluded that there were significant differences in soil C, N, P, enzyme activity and their ratio characteristics under different grassland uses. EG is closer to the standard ratio of global soil ecosystems. Therefore, reducing soil disturbance and optimizing fertilization are effective ways to improve soil enzyme activity and maintain good soil nutrient circulation.

**Keywords:** nutrient limitation; artificial grassland; sustainable utilization; ecological restoration; karst; redundancy analysis





## 1. Introduction

Soil nutrients such as carbon (C), nitrogen (N) and phosphorus (P) are important life elements, which determine the level of soil fertility and directly affect the growth and development of plants. The three elements achieve material circulation and energy flow between soil and plants through migration and transformation in the ecosystem [1,2]. Soil C, N, P ratio can be used to judge the decomposition of soil organic matter, nutrient restriction, C, N, P saturation and other conditions, and it is an important indicator in soil ecological stoichiometry [3]. Utilization patterns affect the primary productivity of grassland ecosystems and change the biogeochemical cycles among key elements such as C, N and P, which is an important driving force for ecosystem processes. Ecological stoichiometry is often used to clarify the relationship between soil and plants in grassland ecosystems, and the ratio of C:N:P is often used to measure the distribution of elements in various components of the ecosystem [4]. Changes in utilization patterns will also lead to changes in soil erosion, soil fertility, biodiversity and ecosystem functions, and then affect soil microbial biomass carbon, microbial biomass nitrogen (MBN), substrate induced respiration (SIR), microbial metabolic entropy ($qCO_2$) and soil enzyme activity [5,6].

Soil enzymes are a catalyst for the formation, decomposition and transformation of soil organic matter, mainly from the secretions of fine plant roots and soil microorganisms [7]. Soil

enzymes are one of the most active organic components of the soil fraction. Soil enzymes have many ecological links with soil C, N and P. They are the main regulators of soil biochemical processes and are involved in all biochemical processes in the soil environment, and are closely related to organic matter decomposition, nutrient cycling, energy transfer and environmental quality [8–10]. The main relevant soil enzymes involved in the transformation of C, N and P are βG, NAG, LAP and ACP [11]. βG is mainly secreted by cellulose-based microorganisms, which can break down cellulose into polysaccharides for their own growth and can indicate the intensity of heterotrophic respiration using organic or inorganic carbon as a substrate in the soil and participate in the carbon cycle [12]. NAG and LAP are involved in the nitrogen cycle and ACP in the phosphorus cycle [13,14]. The study of soil enzyme activity is extremely important for the evaluation of soil condition and ecological effects during vegetation use, and the four soil enzymes mentioned above are involved in the catalytic reactions of C, N and P soils and can indicate the metabolic levels of soil nutrients. Therefore, the measurement of soil enzyme activity associated with key soil nutrients (C, N and P) is widely used as a potential indicator to determine the impact of land use change and management practices on soil quality and fertility [15]. Previous studies have shown that differences in land use patterns and other factors can significantly affect soil enzyme activity, which suggests that soil enzymes can indicate and reflect the trends and intensity of soil biochemical processes under different human disturbances [16–19]. Therefore, an in-depth investigation of soil enzyme activity is important for predicting changes in soil ecosystem function.

The karst of southern China is one of the three major distribution areas of karst in the world, and its special above-ground andunder-ground "binary" structure causes the overall ecological environment of the region to be fragile, which is characterized by slow soil formation, serious soil erosion and prominent rock desertification under the disturbance of natural and human activities [20]. Rocky desertification is a serious threat to the ecological security and sustainable socio-economic development of the regions [21]. A large number of studies have indicated that the artificial grassland establishment is an important component of the comprehensive rocky desertification prevention and control project, and is an important measure to rapidly repair the damaged ecological environment of rocky desertification [22], which is of great significance to promote ecological reconstruction and economic development. The most important utilization of artificial grasslands is grazing and mowing, and differences in use patterns may alter the nutrient relationships of soil enzymes and key nutrients that have evolved over time [23], with far-reaching effects on ecosystem diversity and function [24]. Although a lot of basic research has been done on the relationship between key nutrient factors and soil enzyme activity in grassland soils [25–29], there is a lack of systematic studies on the rate of carbon, nitrogen and phosphorus and enzyme activity of grassland soils in karst desertification areas, especially for artificial grasslands, which are still at the stage of sporadic exploration. Since 1999, a large-scale special project in the karst areas of southern China has been implemented, which involves returning farmland to forest and grass and the vegetation restoration of rocky desertification management. It is estimated that more than 10,000 km$^2$ of land has been treated by ecological measures dominated by artificial grassland establishment [30]. It is urgent to clarify the mechanisms of carbon, nitrogen, and phosphorus ratio and enzymatic activity in response to environmental factors in grasslands under different utilization patterns. Furthering the understanding of soil C, N, P content distribution and their ratio characteristics of different grassland uses is helpful to identify soil ecological processes and their driving mechanisms, and is of great significance for further improving soil fertility and land use efficiency. Therefore, we propose a scientific hypothesis: Is there an effect of different utilization practices on the ratio characteristics of soil carbon, nitrogen and phosphorus and enzymatic activity in the karst rocky desertification? To verify the above conjecture, the artificial grassland of karst desertification control with different grassland utilization patterns (i.e., the grazing grassland (GG), mowing grassland (MG) and enclosed grassland (EG)) was taken as the research object. The main research contents were as follows: (1) soil nutrients and their ratio characteristics; (2) soil enzyme activities and their ratio charac-

teristics; (3) the relationship between soil enzyme activities and soil nutrients; (4) analysis of the factors affecting soil enzyme activities, so as to provide theoretical support for the sustainable use of artificial grassland and ecological restoration of rock desertification in karst areas of southern China.

## 2. Materials and Methods

### 2.1. Research Area

The study area is located in Salaxi Town, Qixingguan District, Bijie City, Guizhou Province, China ($105°02'01''$–$105°08'09''$ E, $27°11'36''$–$27°16'51''$ N), which is a typical karst plateau mountainous area with light–moderate rocky desertification, with a rocky desertification area of $55.931$ km$^2$, accounting for $64.93\%$ of the total area of the demonstration area. The study area has a humid subtropical monsoon climate, with simultaneous rain and heat, an average altitude of approximately 1800 m, an average annual temperature of approximately 12 °C, a frost-free period of 245 d, an average annual number of sunshine hours of 1360 h, and an average annual rainfall of 984.4 mm, with precipitation concentrated in June to September. The soil is zoned calcareous. The vegetation is predominantly composed of *Cyclobalanopsis glauca*, *Pyracantha fortuneana*, *Rhododendron simsii*, *Juglans regia*, *Rosa roxburghii*, *Artemisia lavandulaefolia*, *Chenopodium glaucum*, *Clinopodium chinense*, *Plantago asiatica*, *Stellaria media*, *Digitaria sanguinalis* and *Polygonum hydropiper*.

### 2.2. Sample Plot Set-Up and Sampling

In order to repair the damaged rocky desertification ecosystem and establish a demonstration area for grassland husbandry for karst rocky desertification management, our team carried out artificial grass planting in the study area in mid-April 2012. The artificial grass is a mixed forage, i.e., *Lolium perenne* + *Dactylis glomerata* + *Trifolium repens*, planted at a seed count of 2:2:1. Since its establishment, the artificial grassland has been used mainly for free grazing, with a stock carrying capacity of sheep unit/600 m$^2$. To reveal the effects of different utilization practices on soil carbon, nitrogen and phosphorus ratio characteristics and enzyme activities, we set up GG and MG in August 2019, with EG as the control treatment, for a comparative study. The grazing livestock were semi-fine wool sheep in Guizhou at the age of about 1 year. Except extreme weather, grazing takes place for approximately 300 days per year. Mowing grassland are left at a stubble height of approximately 5 cm and are mowed according to the normal grazing phenology or when they reach mowing height. Enclosed grassland is not used in any way.

In mid-August 2021, 15 sampling points (3 cm from the base of the plant) were evenly distributed in each sample plot using the "S" shaped multi-point sampling method. After removing the litter layer from the soil surface, soil samples were collected from the surface layer (0–10 cm) using a soil auger, and to reduce spatial heterogeneity, the soil samples from the 15 sampling points were mixed into one sample to obtain a total of 9 soil samples. The soil samples were cleaned of impurities and divided into two parts. One part of the sample was placed in a sealed plastic bag and brought back to the laboratory for the determination of soil enzyme activities. The other part of the soil sample was air-dried indoors and then passed through a 2 mm sieve for soil property determination.

### 2.3. Determination of Soil Properties

In this study, eight soil chemical indicators were measured: pH, soil organic carbon (SOC, g/kg), total nitrogen (TN, g/kg), total phosphorus (TP, g/kg), alkaline nitrogen (AN, mg/kg), nitrate nitrogen (NO$_3^-$-N, mg/kg), ammonium nitrogen (NH$_4^+$-N, mg/kg) and available phosphorus (AP, mg/kg). Four important soil enzyme indicators involved in the C, N and P cycles are βG, for C-acquiring enzymes, NAG and LAP for N-acquiring enzymes, and LCP for phosphorus-acquiring enzymes [31–33].

Soil pH was determined using a pH meter in a suspension with a soil to water ratio of 2.5:1 (PHC-3C, Leimi, Shanghai, China). SOC, TN, TP, AN and AP were determined according to the method described by Li et al. [31]. Of these, SOC and TN were determined

using a fully automated elemental analyser (FlashSmart, Thermo Fisher, Waltham, WA, USA) and TP, AN and AP were determined using a continuous flow analyser (FLOWSYS, SYSTEA, Italy). $NO_3^-$-N and $NH_4^+$-N in soil were extracted with KCl solution according to ISO standards and determined using a continuous flow analyser (FLOWSYS, SYSTEA, Anagni, Italy). The activities of βG, NAG, LCP and LAP in soil enzyme activities were analysed by referring to the method of Jiao et al. [34–36] and determined using an Ultraviolet visible spectrophotometer (Specord 200 PLUS, Analytik, Jena, Germany). The unit of activity was expressed as the unit of $IUg^{-1}$.

### 2.4. Data Processing

Enzymatic stoichiometry is an important indicator to reveal the growth and metabolic processes of microorganisms [37] and to evaluate the limitation status of soil nutrient resources [38]. Using the method of Sinsabaugh et al. [32] and Wang et al. [33], the enzyme ratio was calculated as follows: soil C:N enzyme activity ratio (C:NEEA) = ln (βG):ln (NAG + LAP); soil C:P enzyme activity ratio (C:PEEA) = ln (βG):ln (ACP); soil N:P enzyme activity ratio (N:PEEA) = ln (NAG + LAP):ln (AP).

Excel 2013 was used for statistical and preliminary analysis of experimental data. One-way ANOVA, LSD multiple comparison and Person correlation analyses were conducted in SPSS 22 to study the effects of different utilization methods on soil nutrients, enzyme activities and their ratio characteristics, and the map was made in Origin 2018. Canoco 5.0 redundancy (RDA) analysis was used to determine the relationship between soil enzyme activity and their ratio and various environmental factors. These indices, measures of the enzymatic resources directed towards the acquisition of organic P and organic N relative to C, were used to test for functional convergence in soil extracellular enzyme activity distributions across ecosystems and to compare relative nutrient demand [32].

## 3. Results

### 3.1. C, N, P and Their Ratio Characteristics of Grassland Soil under Different Utilization Methods

Different utilization methods had different effects on soil C, N and P contents and their ratios of artificial grassland (Table 1). Compared with EG, GG significantly changed soil pH and increased soil TN and AN contents ($p < 0.05$), but significantly decreased soil TP content ($p < 0.05$). MG significantly decreased the contents of SOC, TP, AN, $NH_4^+$-N and $NO_3^-$-N ($p < 0.05$). GG and MG had no significant effect on AP ($p > 0.05$).

**Table 1.** Contents and ratios of C, N and P in grassland soil under different utilization methods.

| Items | Utilization Methods | | |
|---|---|---|---|
| | GG | MG | EG |
| pH | 7.01 ± 0.06a | 6.78 ± 0.09ab | 6.53 ± 0.20b |
| SOC (g/kg) | 19.48 ± 1.12ab | 17.37 ± 0.42b | 22.45 ± 1.39a |
| TN (g/kg) | 2.19 ± 0.09a | 1.74 ± 0.10bc | 1.83 ± 0.09b |
| TP (g/kg) | 0.85 ± 0.05b | 0.53 ± 0.01c | 1.26 ± 0.020a |
| AN (mg/kg) | 118.49 ± 1.00a | 93.28 ± 1.41c | 104.62 ± 1.26b |
| AP (mg/kg) | 12.00 ± 1.01a | 12.25 ± 1.36a | 15.97 ± 2.12a |
| $NH_4^+$-N (mg/kg) | 34.34 ± 3.08a | 18.03 ± 0.85b | 35.94 ± 1.13a |
| $NO_3^-$-N (mg/kg) | 98.25 ± 3.82a | 79.9 ± 5.07b | 87.72 ± 6.99a |
| C:N | 8.91 ± 0.97c | 10.04 ± 0.67b | 12.31 ± 1.91a |
| C:P | 22.78 ± 0.83b | 32.88 ± 0.79a | 17.91 ± 2.07c |
| N:P | 2.58 ± 0.36b | 3.29 ± 0.27a | 1.46 ± 0.08c |

Mean value (mean ± standard deviation, n = 3). Different lowercase letters indicate the difference between treatments reaching a significant level ($p < 0.05$).

The variation range of soil pH under different utilization methods was from 6.53 to 7.01, showing the characteristics of GG > MG > EG, with significant difference between GG and EG ($p < 0.05$). SOC varied from 17.37 to 22.45 g/kg, showing the characteristics of EG > GG > MG, with significant difference between MG and EG ($p < 0.05$). TN varied

greatly, ranging from 1.74 to 2.19 g/kg, and showed the characteristics of GG > EG > MG, with significant difference between GG and MG ($p < 0.05$). TP varied greatly, ranging from 0.53 to 1.26 g/kg, showing the characteristics of EG > GG > MG, and there were significant differences among different utilization methods ($p < 0.05$). AN was significantly different, ranging from 93.28 to 118.49 g/kg, showing the characteristics of GG > EG > MG, and there was a significant difference between GG and MG ($p < 0.05$). There was little difference in AP, ranging from 12.00 to 15.97 mg/kg, showing the characteristics of EG > MG > GG, and there was no significant difference among different utilization methods ($p < 0.05$). $NH_4^+$-N was significantly different, ranging from 18.03 to 35.94 mg/kg, showing the characteristics of EG > GG > MG, and there was a significant difference between MG and other treatments ($p < 0.05$). There was a significant difference in $NO_3^-$-N, ranging from 79.9 to 98.25 mg/kg, showing the characteristics of GG > EG> MG, and there was a significant difference between MG and other treatments ($p < 0.05$).

C:N was significantly different, ranging from 8.91 to 12.31, showing the characteristics of EG > MG > GG, and there were significant differences among different utilization methods ($p < 0.05$). C:P was significantly different with a range of 17.91–32.88, showing the characteristics of MG > GG > EG, and there was a significant difference between MG and EG ($p < 0.05$). N:P was significantly different, ranging from 1.46 to 3.29, showing the characteristics of MG > GG > EG, and there were significant differences among different utilization methods ($p < 0.05$).

### 3.2. Soil Enzyme Activity and Its Ratios of Grassland under Different Utilization Methods

Different utilization methods had different effects on enzyme activities and their ratios in grassland soil (Table 2). Compared with EG, MG significantly decreased the contents of βG and NAG in soil ($p < 0.05$). GG and MG had no significant effects on ACP and LAP ($p > 0.05$).

**Table 2.** Soil enzyme activity and its ratio under different utilization methods.

| Items | Utilization Methods | | |
|---|---|---|---|
| | GG | MG | EG |
| βG (IUg$^{-1}$) | 6.11 ± 0.05a | 3.54 ± 0.05b | 5.36 ± 0.18a |
| NAG (IUg$^{-1}$) | 8.56 ± 0.11a | 5.36 ± 0.18b | 7.19 ± 0.14a |
| ACP (IUg$^{-1}$) | 6.79 ± 0.72a | 7.46 ± 0.84a | 7.66 ± 0.59a |
| LAP (IUg$^{-1}$) | 29.83 ± 4.28a | 31.27 ± 3.73a | 28.52 ± 3.22a |
| C:NEEA | 0.50 ± 0.03a | 0.35 ± 0.02b | 0.47 ± 0.03a |
| C:PEEA | 0.96 ± 0.09a | 0.64 ± 0.08b | 0.83 ± 0.06a |
| N:PEEA | 1.93 ± 0.28a | 1.82 ± 0.25a | 1.76 ± 0.04a |

Mean value (mean ± standard deviation, n = 3). Different lower case letters indicate the difference between treatments reaching a significant level ($p < 0.05$).

βG ranged from 3.54–6.11 IUg$^{-1}$, showing a GG > EG > MG distribution, with significant differences between MGand other treatments($p < 0.05$), but not significant differences between GG and EG ($p > 0.05$). The variation range of NAG was 5.36–8.56 IUg$^{-1}$, which also showed the distribution characteristics of GG > EG > MG. There were significant differences between MG and other treatments ($p < 0.05$), but no significant differences between GG and EG ($p > 0.05$). ACP ranged from 6.79–7.66 IUg$^{-1}$, showing the distribution characteristics of EG > MG > GG, and there was no significant difference among treatments ($p > 0.05$). The variation range of LAP was 28.52–31.27 IUg$^{-1}$, showing the distribution characteristics of MG > EG > GG, and there was no significant difference among treatments ($p > 0.05$).

The range of variation of C:NEEA was 0.35–0.50, showing GG > EG > MG, with significant differences between MG and other treatments ($p < 0.05$). The range of variation of C:PEEA was 0.64–0.96, showing GG > EG > MG, with significant differences between MG and other treatments ($p < 0.05$). The range of variation of N:PEEA was 1.76–1.93, showing GG > MG > EG, with no significant differences between treatments ($p > 0.05$).

### 3.3. Overall Correlation Analysis of Soil Enzymes, Their Ratios and Soil Chemical Factors in Different Utilization Methods of Grassland

The Figure 1 showed that there was a significant negative correlation between pH and AP ($p < 0.01$), but no significant correlation between other factors. SOC was significantly positively correlated with TP, $NH_4^+$-N, C:N ($p < 0.05$), significantly negatively correlated with C:P, N:P ($p < 0.05$). TN was significantly positively correlated with AN, NAG, βG, C:NEEA ($p < 0.05$). TP showed a significant positive correlation with $NH_4^+$-N and C:NEEA ($p < 0.05$), and significant negative correlation with C:P, N:P ($p < 0.01$). AN showed highly significant positive correlations with βG, NAG, C:NEEA and C:PEEA ($p < 0.01$) and significant positive correlations with $NH_4^+$-N and $NO_3^-$-N ($p < 0.05$). $NH_4^+$-N was positively correlated with βG, NAG, C:NEEA, C:PEEA ($p < 0.05$), negatively correlated with C:P, N:P ($p < 0.01$). C:N was significantly negatively correlated with N:P ($p < 0.05$). C:P was significantly positively correlated with N:P ($p < 0.01$), negatively correlated with NAG, βG, C:NEEA, C:PEEA ($p < 0.05$), but not significantly correlated with other factors. NAG was significantly positively correlated with βG, C:NEEA, C:PEEA ($p < 0.001$). βG was significantly positively correlated with C:NEEA, C:PEEA ($p < 0.01$). ACP was significantly negative correlated with ACP, and C:NEEA were significantly positively correlated with C:PEEA ($p < 0.01$).

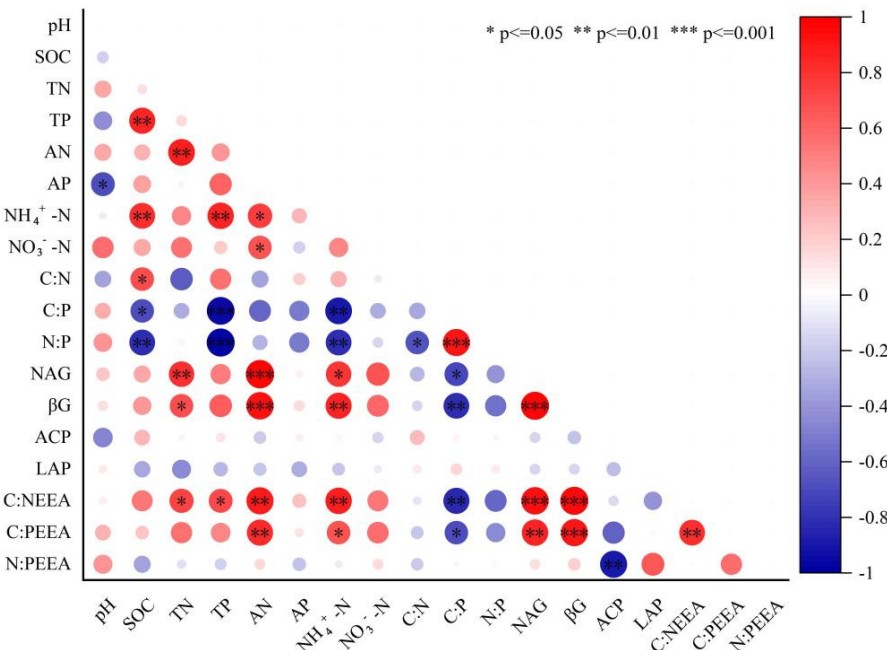

**Figure 1.** Correlation analysis of soil nutrients and soil enzyme activity in grasslands with different use patterns.

### 3.4. Analysis of Soil Chemical Factors Affecting Enzyme Activity

Forward-selection-based redundancy analysis (RDA) was used to select explanatory environmental factors that had significant effects on soil enzyme activities ($p < 0.05$). Soil enzyme activity was taken as a biological variable and soil physical and chemical factors as environmental variable, and then constraint treatment was carried out between biological and environmental variables in standard analysis to obtain the different explanatory quantity of soil physical and chemical factors on soil enzyme activity. According to the redundancy analysis of soil activity, their ratio and soil properties showed that the first and second axes of soil environmental factors explained 53.21% and 36.50% of the variation of soil enzyme activity, respectively (Figure 2). In conclusion, the first two axes can well reflect the relationship between soil enzyme activity and soil environmental factors, and are mainly determined by the first axis. Further analysis of the Monte Carlo replacement

test showed that AN (F = 4.90, *p* = 0.01) was more correlated with soil activity and its ratio than other soil environmental factors, and was the main factor affecting soil activity and its ratio. βG, C:NEEA, C:PEEA were positively correlated with all environmental factors except C:P and N:P, LAP was positively correlated with C:P and N:P and negatively correlated with other environmental factors. ACP was positively correlated with SOC, C:P, N:P and negatively correlated with other soil environmental factors. N:PEEA was negatively correlated with SOC.

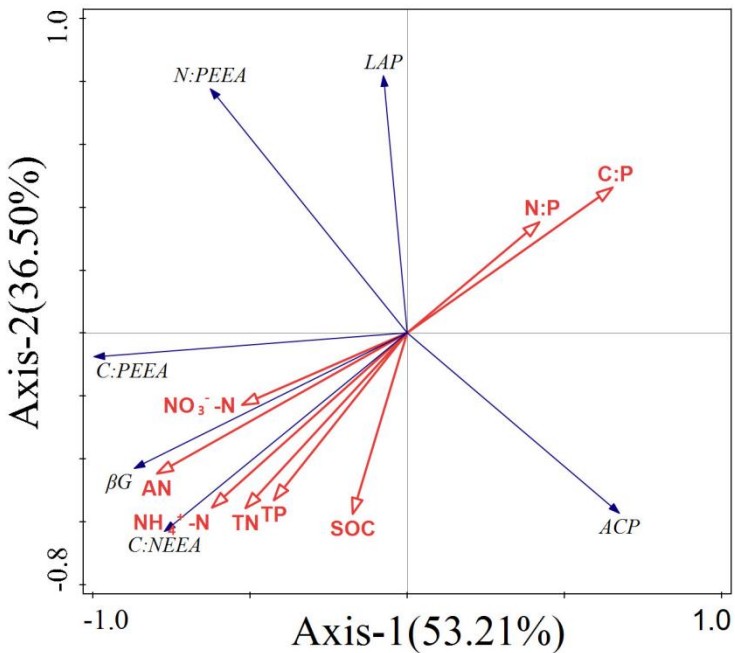

**Figure 2.** RDA analysis of soil enzyme activity and soil chemical factors.

## 4. Discussion

### 4.1. Characteristics of Soil C, N and P Nutrients

C, N and P are the three most important living elements on the earth. There is a stable rate relationship between the demands of C, N and P elements by organisms. The ratio of C, N and P in the soil environment determines the nutrient availability for plants and microorganisms, thereby reflecting the ecosystem's functionality [39]. There were some differences in the soil nutrient characteristics of three different treatments of artificial grassland. In this study, the average SOC content of grassland under GG, MG and EG utilization was 19.48 g/kg, 17.37 g/kg and 22.45 g/kg, respectively, indicating that EG utilization may be beneficial to SOC accumulation, followed by GG and MG. In this study, the soil SOC content of the three grassland utilization methods was at the medium level of the grassland SOC content in the rocky desertification area. The soil SOC content in this study was higher than the 9.99 g/kg of Guo et al. [6] and lower than the 36.22 g/kg of Tian et al. [40], but close to the 18.12 g/kg of the previous research results of the research group [41]. However, there is still a certain gap between the SOC content of 105.10 g/kg in forest soil [42].

Soil nitrogen is the largest mineral element absorbed by plants from soil and the main limiting factor for vegetation growth in grassland [43]. In this study, TN content in GG, MG and EG treatments was 2.19 g/kg, 1.74 g/kg and 1.83 g/kg, respectively, and TN content under GG treatment was the highest, which also indicated that the manure of grazing livestock had a promoting effect on the improvement of TN in grassland soil to a certain extent, which was consistent with the previous results of the research group [41,44]. However, there is still a certain gap with the average content (3.24 g/kg) of similar studies in the karst mountain area of Ziyun County, Guizhou Province [45]. Although soil inorganic nitrogen such as $NO_3^-$-

$N/NH_4^+$-N only accounts for a small part of soil TN, this part of nitrogen is the nitrogen source directly absorbed and utilized by plants and is the most effective soil nitrogen index to evaluate soil productivity level. Under the action of microorganisms, it can be directly absorbed and utilized by plants to change soil properties [43]. In this study, the content of AN and $NO_3^-$-N in GG treatment were higher than that in MG treatment, while EG was at the median level. For $NH_4^+$-N, EG was the highest, GG was slightly lower than EG, and MG was the lowest, indicating that soil N's available state in GG was relatively high, EG was in the middle and MG was the lowest. The average TP content was 0.85 g/kg, 0.53 g/kg and 1.26 g/kg, respectively, and the highest TP content was obtained under EG treatment, which also indicated that soil TP improvement was promoted by enclosure to a certain extent, which was consistent with the study in the desert steppe area of northwest China. Moreover, soil N storage increased significantly in soil <20 cm [46].

In this study, the content of AP in EG was the highest, while MG and GG were slightly lower, indicating that compared with human disturbance treatment, EG may increase the content of AP through the backtracking of litter. This is somewhat different from the study of Lu et al. in the alpine grassland of Tibet. Lu believed that there was no difference in soil's available phosphorus content between the fenced grassland and the unfenced grassland after 6–8 years of fenced experiment, which may be related to the fenced time [47]. With the increase in fenced years, the adverse effects of fencing become increasingly prominent [48].

Soil pH is an important factor affecting grassland soil quality, and fencing can significantly reduce soil pH and improve soil salinization [49]. With the removal of anthropogenic stress such as grazing, aboveground biomass, litter biomass and root biomass increased, and microbial metabolism in the rhizosphere became more active. As a result, the root system secreted a large amount of organic acids, and $CO_2$ released by the root system and soil microorganisms increased, which led to the decrease in pH of grassland soil [50]. In addition, the results of this study also confirmed that compared with GG and MG treatments, EG treatment had the fastest pH decline, resulting in soil acidification. The research results were consistent with previous studies, and fencing resulted in significant acidification of grassland soil, especially the largest pH decline rate of surface soil [51].

Therefore, studies have shown that biological and abiotic factors related to the decomposition of soil organic matter and nutrient cycling in the ecosystem jointly determine the changes in soil C, N and P contents [52]. The results of this study also indicated that different anthropogenic disturbances, namely different utilization methods, increased the heterogeneity of the grassland soil environment, and ultimately affected the changes of soil nutrient elements such as C, N, P and pH.

### 4.2. Ratio Characteristics of Soil C, N and P

Soil C, N, P ratio is an important index to measure soil organic matter composition and nutrient balance and can indirectly represent the rate of nutrient return to soil [53]. C:N is an important indicator of soil N mineralization ability, which determines the decomposition rate of organic matter [54] and can determine whether mineralization or microbial retention occurs during the decomposition process [55]. In this study, C:N in GG, MG and EG were 8.91, 10.04 and 12.31, respectively, and EG was more similar to the value of C:N of surface soil in China, which was found by Tian et al. [56]. Studies have also shown that when C:N is less than 25, the microbial decomposition activity is enhanced, and the decomposition and mineralization rate of soil organic matter is accelerated [57], which also indicates that the enclosed grassland may have strong microbial decomposition ability and high mineralization rate, which is more conducive to the decomposition of soil organic matter, but not conducive to the accumulation of organic matter.

When C:P is greater than 300, soil nutrient retention is net; when C:P is less than 200, microbial P element is net mineralized; when C:P is 200–300, soluble P concentration changes little [58]. In this study, the C:P treated by GG, MG and EG were 22.78, 32.88 and 17.91, respectively, much lower than the average value of 136 in China [56], which also

indicated that soil microorganisms in the study area had great potential to release P in mineralized soil organic matter.

Previous studies have pointed out that plants are in a N-restricted state when N:P is less than 10 [59,60]. In this study, the N:P values of GG, MG and EG treatments were 2.58, 3.29 and 1.46, respectively, much lower than the average value of 9.3 in China, indicating that soil nutrients of the three grassland use patterns in the study area were mainly restricted by N. This is consistent with the research of Wu et al. [61], Qi et al. [11] and Guo et al. [6], but inconsistent with the research results of Tian et al. [40] and Liu et al. [62] that soil nutrients in karst areas are mainly limited by P. This may be mainly due to the existence of a large amount of litter and microorganisms on the soil surface under different utilization methods, and the N element mainly comes from the decomposition of litters by microorganisms, which ultimately leads to certain differences in microbial activity and soil organic matter content among different utilization methods. Compared with GG and MG, EG is more significantly limited by N. According to the growth rate hypothesis (GRH), the biomass N/P of plants growing rapidly is usually very low [63]. Due to the absence of human disturbance, the faster growth rate of EG grassland led to an increase in the demand for the limiting element N, which prompted the herbage to strengthen the absorption of N from the soil, thus leading to a rapid decline in N:P values.

Therefore, in order to realize the sustainable utilization of artificial grassland, the rational application of N fertilizer and exogenous N input should be considered from the perspective of soil nutrient limiting elements, so as to effectively alleviate the restriction of N on soil nutrients and promote the soil nutrient balance [6,64]. The study of soil C, N and P balance relationship is of great significance for karst ecological restoration.

### 4.3. Responses of Soil Enzymes to Different Grassland Uses

As one of the important indicators of soil fertility, soil enzyme activity is directly related to the function of the soil ecosystem, influenced by soil microorganisms, fine root secretions, litter decomposition, soil animals and their residues' decomposition, etc. [65]. Different land use types change plant functional groups, species richness, hydrothermal conditions, nutrient sources, profile structure and nutrient cycling in the most direct and rapid way of human activities, leading to changes in soil enzyme activity and fertility characteristics [66,67]. In this study, there were significant differences in soil enzyme activity among different grassland utilization methods. According to the theory of resource allocation, when soil nutrient deficit occurs, microorganisms secrete related enzymes to improve the utilization of limiting elements, thus effectively alleviating the restriction of soil nutrients [68]. In this study, it was found that the βG and NAG enzyme activities of GG treatment were higher than those of EG treatment, and significantly higher than those of MG treatment, which may be due to the relatively high demand of soil microorganisms of GG treatment for C and N nutrient elements, which was consistent with the results of Xu et al. [69]. The enzyme activities of GG and NAG treatments were significantly higher than those of EG treatments, mainly because livestock grazed freely and excreta released a certain amount of N, which directly increased soil TN, led to the decrease of soil C:N and promoted the decomposition and cyclic metabolism of SOC and N. The ACP activity of EG treatment was significantly higher than that of GG treatment, which may be due to the increase of soil nitrogen caused by grazing, thus improving the ACP activity, which is consistent with the results of Tian et al. [70], indicating that higher nitrogen level leads to a significant increase in ACP activity. ACP is mainly derived from fungi and plant roots, and the aboveground biomass and leaf phosphorus content of grazing grassland plants are significantly increased under the relatively high nitrogen input induced by grazing and the ACP activity of plant roots is enhanced [71,72].

### 4.4. Effects of Utilization Methods on Soil Enzyme Ratio

The ratio of enzymes involved in C, N and P cycles is used to characterize the restriction of energy (C) and nutrients (N, P) in the growth and metabolism of soil microorganisms [73].

According to the study of Sinsabaugh et al., the quantitive ratio of chemical enzymes in global ecosystem was (C:NEEA):(C:PEEA):(N:PEEA) = 1:1:1, while in this study, the (C:NEEA):(C:PEEA):(N:PEEA) of GG, MG and EG were 1.41:1:1.33, respectively. 1.5:1:1.3 and 1.1:1.03:1, all deviated from the standard ratios of global soil ecosystems [74]. However, in this study, EG treatment is relatively close to GG, MG and the standard ratio of the global ecosystem. These results indicated that GG and MG treatments had a higher demand for C and P, while EG treatments had a relatively higher demand for C and N, indicating that soil enzyme activities in different utilization methods of artificial grassland may be limited by C, N and P to a certain extent, but the degree of restriction is different. This is basically consistent with the research done by Xu et al. in the Guizhou karst area [69]. In this study, soil enzyme C:NEEA value ranged from 0.35 to 0.50, lower than the global average (1.41); soil enzyme C:PEEA value ranged from 0.64 to 0.96, higher than the global average (0.62); soil enzyme N:PEEA value ranged from 1.76 to 1.93. It is higher than the global average (0.44) [15]. These results indicated that soil enzyme activities of the three utilization methods had higher participation in C and N metabolic activities. According to the "optimal allocation" principle of resource allocation theory, microorganisms release more enzymes corresponding to lack of resources, thus balancing the nutrient and energy constraints [75]. The results indicated that the grasslands with different utilization methods were subject to different degrees of C and N restriction, and the soils treated with GG and MG were subject to greater degrees of C and N restriction. This also reflects the fact that soil enzyme activity and its ratio depend to some extent on vegetation use patterns.

*4.5. Soil Enzymes and Environmental Factors*

Soil enzyme activity and soil nutrients have a typical reciprocal relationship, mutual influence and interaction. In this study, RDA analysis showed that the main soil chemical index factor affecting enzyme activity was soil AN, which was consistent with the results of previous studies, and it was believed that AN played an important driving role in the change of soil enzyme activity [45,76]. Studies have also shown that in addition to TN, SOC and TP are also important factors affecting soil enzymes [77]. It can be seen that soil enzyme activity is subject to the superimposed effect of multiple soil physical and chemical factors, and different utilization methods lead to different driving factors affecting soil enzyme activity among different treatments. In addition, soil moisture content and microbial quantity [78,79], which were not involved in this study, may also be important factors affecting soil enzymes, and further research should be strengthened in the future.

Soil C, N, P, C:N, C:P and N:P jointly affect the growth and development of soil microorganisms, including soil microbial biomass carbon, and ultimately affect the change of soil enzyme activities [80]. It has been shown that soil enzymes can increase the input of soil nutrients by enhancing plant biomass and fine root secretions as well as promoting microbial carbon production, while soil microorganisms, after acquiring nutrients and energy, can regulate enzyme secretion and change the biochemical properties of enzymes, thereby increasing soil enzyme activity, according to the resource allocation theory [81]. At the same time, the synthesis of soil enzymes requires a certain amount of energy to be involved [82]. When soil nutrients are limited by N, more SOC and TP will be used to synthesize energy for limiting nutrient (N) enzymes in order to meet the normal supply of N in relatively resource-poor areas, which may also be one of the reasons why N is one of the main factors affecting the enzyme activity in southwest karst soils. It has been shown that with increasing intensity of human disturbance, it is possible to cause different degrees of reduction in soil enzyme activity [83,84]. Thus, decreasing the intensity of soil disturbance is a way to increase soil enzyme activity and maintain good soil nutrient cycling, which was confirmed by the enzyme activity and its enzyme ratio in this study.

## 5. Conclusions

This study was based on an soil ecosystem of artificial grassland in karst desertification management area in southern China. We explored the effects of different uses on soil

chemical properties and soil enzyme activity in artificial grassland. We conclude that: (1) there are significant differences in soil C, N, P and their rate characteristics between grassland use patterns. The highest levels of SOC and TP were found in the EG treatment, while TN levels were highest in the GG. C:N < 12.3 was found in the GG and MG treatments, and soil N:P < 9.3 was found in all three treatments, which indicates that soil nutrients were N-limited in the EG treatment, while soil nutrients were N and P-limited under the GG and MG treatments; (2) There were significant differences in soil enzyme activities under different grassland utilization. The activity of βG and NAG was highest in GG, ACP in EG and LAP in MG; (3) AN is an important driver of changes in soil enzyme activity; (4) Compared to GG and MG, the soil enzyme ratio (C:NEEA):(C:PEEA):(N:PEEA) of EG is 11.1:1.03:1, which is close to the standard ratio for global soil ecosystems. This suggests that reducing soil disturbance in grassland (e.g., timely enclosure) and optimizing fertilizer application are effective ways to improve soil enzyme activity and maintain good soil nutrient cycling. In this study, different grassland use patterns significantly changed the soil chemical properties and their ratio characteristics as well as soil enzyme activities and their ratio characteristics. In future studies, it is necessary to continue long-term observations and add suitable environmental factors, so as to reveal the mechanisms of long-term different use patterns on the soil environmental system and serve for the restoration of the rocky desertification ecosystem and sustainable grassland use.

**Author Contributions:** Conceptualization, Y.C. and S.S.; methodology, X.W. and C.H.; software, X.W. and C.H.; validation, Y.C., S.S. and X.W.; formal analysis, X.W.; investigation, X.W.; resources, Y.C.; data curation, X.W.; writing—original draft preparation, Y.C.; writing—review and editing, S.S.; visualization, S.S.; supervision, Y.C. and C.H.; project administration, Y.C.; funding acquisition, Y.C. All authors have read and agreed to the published version of the manuscript.

**Funding:** This research was funded by the Natural Science Research Project of Education Department of Guizhou Province [Qianjiaohe KY Zi (2022) 157]; Academic New Seedling Fund Project of Guizhou Normal University (Qianshi Xinmiao B15); Guizhou Province Graduate Education Innovation Program (Qianjiaohe YJSKYJJ[2021]097).

**Data Availability Statement:** Not applicable.

**Acknowledgments:** The authors thank the reviewers and editor for their insightful comments and constructive suggestions.

**Conflicts of Interest:** The authors declare no conflict of interest.

## Abbreviations

| | |
|---|---|
| GG | grazing grassland |
| MG | mowing grassland |
| EG | enclosed grassland |
| pH | pondus hydrogenii |
| SOC | soil organic carbon |
| TN | total nitrogen |
| TP | total phosphorus |
| AP | soil available phosphorus |
| AN | alkaline nitrogen |
| $NH_4^+$-N | soil ammonium nitrogen |
| $NO_3^-$-N | soil nitrate nitrogen |
| βG | β-1,4-glucosidase |
| NAG | β-1,4-N-acetylglucos-aminidase |
| LAP | leucine aminopeptidase |
| ACP | acid phosphatase |

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
