# Peer review of "Effects of Utilization Methods on C, N, P Rate and Enzyme Activity of Artificial Grassland in Karst Desertification Area"

_agronomy, doi:10.3390/agronomy13051368_

Round 1

Reviewer 1 Report

Dear Author,

The manuscript is interesting but has shortcomings.

1. Explain in the introduction the differences between GG - grazing grassland, MG - mowing grassland and EG - enclosed grassland.

2. Chapter 2.2. Describe the methodology for the determination of enzymes. The indicated scientific article is not available to a wide range of readers.

3. Chapter 2.2.How did you mark AN? What indicator?

4. Chapter 2.3. For what purpose did you enumerate C:PEEA, C:NEEA, N:PEEA? Explain what are indicators? What do they inform about? Why did you choose them? Relevant information is missing. However, item [33] is not available to readers.

5. Chapter 4.4. Explain what "global ecosystems" mean? Which ecosytenes are referred to by the authors of article no. 70? Neither article 32 nor article 70 contained information (L.: 163-166, 406-410). In my opinion there is a lack of integrity here.

Good luck.

Reviewer

Author Response

Dear editor and Reviewers,

Thank you for your letter and the reviewers’ comments concerning our manuscript entitled “Effects of Utilization Methods on Stoichiometric Characteristics of Soil C, N, P and Enzyme Activities of Artificial Grassland in Karst Rocky Desertification Area” (ID: 2371280). Those comments are valuable and very helpful. We have read through comments carefully and have made corrections. Based on the instructions provided in your letter, we uploaded the file of the revised manuscript. Revisions in the text are shown using red highlight. The responses to the reviewer’s comments are presented following. We would love to thank you for allowing us to resubmit a revised copy of the manuscript and we highly appreciate your time and consideration.

Best wishes,

Authors

Reviewer 1

The manuscript is interesting but has shortcomings.

  1. Explain in the introduction the differences between GG - grazing grassland, MG - mowing grassland and EG - enclosed grassland.

Answer: We are grateful for the suggestion. To be more clearly and in accordance with the reviewer concerns, we have added a more detailed interpretation in line 100-107.

  1. Chapter 2.2. Describe the methodology for the determination of enzymes. The indicated scientific article is not available to a wide range of readers.

Answer: Thank you for your comments. We have mainly combined three references to describe the methods for determining enzymes, but the older literature may have been omitted when writing the article, so we added the related references as follows: 

Saiya-Cork, K.R.; Sinsabaugh, R.L.; Zak, D.R. The effects of long term nitrogen deposition on extracellular enzyme activity in an Acer saccharum forest soil. Soil Biol. Biochem. 2002, 34, 1309-1315.

Guan, S. Y.; Zhang, D. S.; Zhang, M.Z. Soil enzymes and their research methods. China Agriculture Press: Beijing, China, 1986; pp. 274–337.

3.Chapter 2.2.How did you mark AN? What indicator?

Answer: Thank you very much for pointing out this problem, our explanation is mainly as follows:

AN is a kind of available nitrogen which can be absorbed by crops during their growth period.  The content of AN depends on the content of organic matter, the quality of organic matter and the amount of nitrogen fertilizer.  The content of organic matter is rich, the maturity is high, the content of AN is high, otherwise the content is low.  The content of AN in soil is not stable and is easily affected by soil moisture and heat conditions and biological activities, but it can reflect the recent nitrogen supply capacity of soil.  The content of AN as a plant nitrogen nutrition has a better correlation than inorganic nitrogen, so it is often used as an indicator of soil nitrogen availability.

4.Chapter 2.3. For what purpose did you enumerate C:PEEA, C:NEEA, N:PEEA? Explain what are indicators? What do they inform about? Why did you choose them? Relevant information is missing. However, item [33] is not available to readers.

Answer: We are extremely grateful to reviewer for pointing out this problem. Our explanation is mainly as follows: Enzymatic stoichiometry is an important indicator to reveal the growth and metabolic processes of microorganisms and to evaluate the limitation status of soil nutrient resources. Ratios of ln(BG) : ln(AP) and ln(BG) : ln(NAG + LAP) activities were calculated for all cases. These indices, measures of the enzymatic resources directed towards acquisition of organic P and organic N relative to C, were used to test for functional convergence in soil extracellular enzyme activity distributions across ecosystems and compare relative nutrient demand.

According to the reviewer’s comment, we have added the relevant information Chapter 2.3, the detailed can be seen in line 169-171, 181-184.

  1. Chapter 4.4. Explain what "global ecosystems" mean? Which ecosystems are referred to by the authors of article no. 70? Neither article 32 nor article 70 contained information (L.: 163-166, 406-410). In my opinion there is a lack of integrity here.

Answer: We are grateful for the suggestion. We apologize for the problems in the original manuscript, we may not have translated it completely. The term “global ecosystems” refers to the global soil ecosystem. Thank you very much for pointing it out. Relevant references are as follows:

Sinsabaugh, R.L.; Hill, B.H.; Shah, J.J.F. Ecoenzymatic stoichiometry of microbial organic nutrient acquisition in soil and sediment. Nature 2009, 462, 795–798.  

Reviewer 2 Report

The study is highly scientific, but it needs a little terminological correction.

I have the following suggestions for the authors to consider:

The title: "Effects of Utilization Methods on Stoichiometric Characteristics of Soil C, N, P and Enzyme Activities of Artificial Grassland in Karst Rocky Desertification Area ", more than 20 is extended.

From my point of view, the terminology "Stoichiometric" is confusing because the authors had interpretations about the mass ratio between elements - C, N, P and enzyme activities.

It's just a suggestion if they accept to change the title: "A long-term study on the change in C, N, P rate and enzyme activity of desertification territory under different artificial grassland".

It is a good practice not to repeat words between the title and keywords to increase search coverage.

The methods must specify whether you use a mass or molar ratio between the elements.

Author Response

Dear editor and Reviewers,

Thank you for your letter and the reviewers’ comments concerning our manuscript entitled “Effects of Utilization Methods on Stoichiometric Characteristics of Soil C, N, P and Enzyme Activities of Artificial Grassland in Karst Rocky Desertification Area” (ID: 2371280). Those comments are valuable and very helpful. We have read through comments carefully and have made corrections. Based on the instructions provided in your letter, we uploaded the file of the revised manuscript. Revisions in the text are shown using red highlight. The responses to the reviewer’s comments are presented following. We would love to thank you for allowing us to resubmit a revised copy of the manuscript and we highly appreciate your time and consideration.

Best wishes,

Authors

Reviewer 2

  1. The study is highly scientific, but it needs a little terminological correction.

I have the following suggestions for the authors to consider:

The title: "Effects of Utilization Methods on Stoichiometric Characteristics of Soil C, N, P and Enzyme Activities of Artificial Grassland in Karst Rocky Desertification Area ", more than 20 is extended.

From my point of view, the terminology "Stoichiometric" is confusing because the authors had interpretations about the mass ratio between elements - C, N, P and enzyme activities.

It's just a suggestion if they accept to change the title: "A long-term study on the change in C, N, P rate and enzyme activity of desertification territory under different artificial grassland".

Answer: We deeply appreciate the reviewer’s suggestion. Because the monitoring time of this paper is only three years, it can not be regarded as long-term monitoring, and in order to fit the research content of this paper, we modify the title to “Effects of Utilization Methods on C, N, P rate and enzyme activity of Artificial Grassland in Karst Desertification Area”.

  1. It is a good practice not to repeat words between the title and keywords to increase search coverage.

Answer: We are extremely grateful to reviewer for pointing out this problem. We have revised the keywords according to your comments. See line 25-26 for details.

3.The methods must specify whether you use a mass or molar ratio between the elements.

Answer: Thank you for your comment. According to your comments, we have specified the “quality ratio” in the method section, see line 152-154 for details.

Round 2

Reviewer 1 Report

I accept the manuscript in this form.

Author Response

Thank you for your letter and comments concerning our manuscript entitled “Effects of Utilization Methods on C, N, P Rate and Enzyme Activity of Artificial Grassland in Karst Desertification Area” (ID: 2371280). Those comments are valuable and very helpful. We have read through comments carefully and have made corrections. Based on the instructions provided in your letter, we uploaded the file of the revised manuscript. Revisions in the text are shown using red highlight. The responses to the editor and reviewer’s comments are presented following. In addition, due to the need of project conclusion, we changed the order of the first author and corresponding author in the revised manuscript, which is hereby explained. We would love to thank you for allowing us to resubmit a revised copy of the manuscript and we highly appreciate your time and consideration.

Best wishes,

Authors

Responses to the editor’s comments 

  1. I agree with the new title suggested by one of the referees. For this reason, the term"stoichiometric" must be deleted/substituted along the text.

Answer: Thank you for your suggestion, we have deleted/substituted the term "stoichiometric" along the text.

  1. In addition, the authors must change: Introduction Nitrogen (N) and phosphorus (P), as the limiting elements.... carbon (C) is not a limiting nutrient! Please amend the text consequently.

Answer: Thank you for your suggestion, we have amended this expression.